# Interventions promoting recovery from depression for patients transitioning from outpatient mental health services to primary care: Protocol for a scoping review

**Anne Sofie Aggestrup**[1]*, **Frederik Martiny**[2,3], **Maria Faurholt-Jepsen**[4], **Morten Hvenegaard**[5], **Robin Christensen**[6], **Annette Sofie Davidsen**[2], **Klaus Martiny**[1]

**1** Copenhagen Affective Disorder Research Centre (CADIC), New Interventions in Depression (NID) Group, Mental Health Centre Copenhagen, Department of Clinical Medicine, University of Copenhagen, Copenhagen, Denmark, **2** The Research Unit for and Section of General Practice, Department of Public Health, University of Copenhagen, Copenhagen, Denmark, **3** Center for Social Medicine, Bispebjerg and Frederiksberg Hospital, Copenhagen, Denmark, **4** Copenhagen Affective Disorder Research Centre (CADIC), Mental Health Centre Copenhagen, Copenhagen University Hospital, Rigshospitalet, Copenhagen, Denmark, **5** Competence Centre for Rehabilitation and Recovery, Mental Health Centre Ballerup, Ballerup, Denmark, **6** Section for Biostatistics and Evidence-Based Research, The Parker Institute, Bispebjerg and Frederiksberg Hospital & Department of Clinical Research, University of Southern Denmark, Odense University Hospital, Denmark

* anne.sofie.aggestrup@regionh.dk

# Abstract

## Introduction

Patients with severe Major Depressive Disorder (MDD) have an increasing risk of new psychiatric hospitalizations following each new episode of depression highlighting the recurrent nature of the disorder. Furthermore, patients are not fully recovered at the end of their treatment in outpatient mental health services, and residual symptoms of depression might explain why patients with MDD have a high risk of relapse. However, evidence of methods to promote recovery after discharge from outpatient mental health services is lacking. The proposed scoping review aims to systematically scope, map and identify the evidence and knowledge gaps on interventions that aims to promote recovery from MDD for patients transitioning from outpatient mental health services to primary care.

## Materials and methods

The proposed scoping review will follow the latest methodological guidance by the Joanna Briggs Institute (JBI) in tandem with the Preferred Reporting Items for Systematic reviews and Meta-Analysis—extension for Scoping Reviews (PRISMA-ScR) checklist. The review is ongoing. Four electronic databases (Medline via PubMed, PsycINFO, CINAHL, and Sociological Abstracts) were systematically searched from 20 January 2022 till 29 March 2022 using keywords and text words. The review team consists of three independent screeners. Two screeners have completed the initial title and abstract screening for all studies retrieved by the search strategy. Currently, we are in the full text screening phase. Reference lists of

**Data Availability Statement:** Data sharing not applicable as no datasets generated and/or analyzed for this study.

**Funding:** This scoping review was funded by Helsefonden (21-B-0478), Jascha Fonden (2021-0082) and the Intersectoral Fund for Health Research (Tværspuljen) in the Capital Region of Denmark (P-2022-1-08). The funding source supports the first authors salary to carry out this review. The funders will not be involved in the study design, data collection, writing the manuscript, and the decision to submit the manuscript for publication.

**Competing interests:** The authors have declared that no competing interests exist.

included studies will be screened, and data will be independently extracted by the review team. Results will be analyzed qualitatively and quantitatively.

## Discussion

The chosen methodology is based on the use of publicly available information and does not require ethical approval. Results will be published in an international peer reviewed scientific journal, at national and international conferences and shared with relevant authorities.

## Registration

A pre-print has been registered at the medRxiv preprint server for health sciences (doi.org/10.1101/2022.10.06.22280499).

## Introduction

### Disease burden of Major Depressive Disorder and treatment across sectors

Major Depressive Disorder (MDD) is one of the most prevalent mental disorders worldwide with a lifetime risk of 20% for adults on a global level [1–3]. Approximately 5% of the general population experiences a depressive episode within a 12-month period [1–3]. Current predictions by the World Health Organization indicate that by 2030 depression will be the leading cause of disease burden globally [4]. MDD negatively impacts quality of life, reduces psychosocial, social, and occupational functioning, and markedly increases morbidity and mortality [5–7].

A large body of evidence from e.g., epidemiological surveys has documented a strong interconnection and increased comorbidity of MDD with other mental disorders, most notably with anxiety disorders and substance use disorders [5, 8–11]. MDD has furthermore been associated with comorbid physical diseases, e.g., diabetes and heart diseases [5, 6, 12–15], and social difficulties [10, 16, 17], e.g., poor work participation, drift to a lower social class, and poorer education.

Thus, MDD is a major burden for the individual patient and public health throughout the world [1, 2, 18, 19], and on a societal level, MDD leads to significant direct costs for treatment, care, and rehabilitation and indirect costs due to disease-related work disability and mortality [5, 6, 12–15].

A diagnosis of MDD is reached when patients experience five or more out of nine symptoms during the same 2-week period and at least one of these symptoms should be either depressed mood or loss of interest and pleasure [20]. In addition, patients with MDD experience a variety of associated emotional, cognitive, and behavioral symptoms [10, 21, 22].

Most patients with MDD are diagnosed and treated in primary care by general practitioners, but research has shown that the ability to detect, diagnose, and treat patients with MDD is often insufficient [23–25]. Furthermore, there is sparse evidence to conclude which type of treatment approach is most effective in preventing relapse or recurrence of MDD [26, 27]. A Cochrane review concluded that patients taking antidepressant medication were less likely to relapse or to experience a recurrent episode compared to patients not taking antidepressant medication (13.9% versus 33.8%) [27]. There are, however, methodological problems in assessing the prophylactic effect of antidepressants [28]. There is also some evidence that non-pharmacological treatment options can induce a reduction in depression symptoms and further remission, i.e., rumination-group cognitive-behavioral therapy, light, exercise, and sleep regulation therapy [29, 30].

MDD is often reoccurring and in some cases it becomes chronic. After treatment of the first episode of severe MDD, more than 50% of all inpatients will relapse [15, 31–36]. These residual symptoms of MDD and incomplete recovery are thus considered a significant contributor to the high risk of relapse for patients with MDD [37]. Therefore, MDD requires long-term and adequate multimodal treatment to induce recovery and prevent/reduce the risk of further episodes. Specialized mental health services typically manage treatment of severe recurrent depression and difficult to treat depression or pressing suicidal ideation [38–40]. However, most mental health services only offer treatment for shorter periods, and research concludes that too early discharge can remove critical support and treatment from vulnerable patients that are not fully recovered at the end of their treatment in outpatient mental health services [36, 41, 42]. In addition, research has shown that patients may not have full confidence in the general practitioners' ability to decide continuation/discontinuation of antidepressants due to a perceived lack of knowledge and time in general practice [38, 43]. As described, some patients relapse into the treatment gaps between outpatient mental health services and often insufficient treatment in primary care. Patients with MDD require ongoing maintenance treatments over the long term to facilitate continued recovery.

## Recovering from Major Depressive Disorder

The concept of recovery was first used in the 1960s, primarily aimed to restore human rights as part of user movements responding to the perceived dominant, and stigmatizing notion of mental illness as chronic with little possibility for improvement [44]. Since then, recovery has become an increasingly important aspect of mental healthcare [45, 46]. The main notions of recovery in mental health literature are the concepts of *clinical*—and *personal* recovery.

Clinical recovery refers to a process of individual recovery from mental illness by remission from symptoms and attainment of functional improvement [47–49]. Personal recovery refers to a process in which the individual recovers from the social consequences of the mental illness, thus regaining a meaningful life and participating in the community by overcoming the challenges of mental illness with or without symptoms. Personal recovery is commonly conceptualized via the 'CHIME' Framework. It consists of five interrelated processes: **C**onnectedness with other people and the community; **H**ope and optimism about the future; overcoming stigma and redefining a positive sense of **I**dentity; **M**eaning in life as defined by rebuilding a meaningful life with social goals; **E**mpowerment, which includes taking personal responsibility and control over one's life [50–57]. The two concepts of recovery have led to some polarization in the understanding of what recovery entails [58]. Recently, it has been argued that the two concepts should be considered complementary rather than contrasting, especially to prevent patients are left in limbo in the, sometimes, polarized discussion between researchers and clinicians [47]. Professionals working in psychiatry tend to focus on clinical recovery [47–49, 58], while general practitioners tend to focus more on personal recovery, which aligns well with a generalist and person-centered rather than a disease-centered approach to care [43, 52, 58]. Yet, patients want both clinical and personal recovery [58]. Therefore, in this review, we focus on both clinical- and personal recovery, collectively referred to as "recovery".

## Existing evidence on recovery from Major Depressive Disorder for patients transitioning from outpatient mental health services to primary care

To our knowledge, there are no published scoping reviews that summarize the evidence for interventions aiming to promote continued recovery from MDD for patients transitioning from outpatient mental health services to primary care. Most studies investigating recovery interventions and/or relapse prevention from MDD have been undertaken in primary care [38, 59, 60]. Two

reviews have a specific focus on developing recovery interventions, e.g., scoping the evidence for internet-based recovery-oriented interventions [61], or developing a proposed logic model, i.e., a visual representation of what works for whom, why, and under which circumstances, for how recovery-oriented interventions could contribute to recovery [62]. However, these reviews were not specific to our target group of interest and did not focus on clinical recovery or patients transitioning from outpatient mental health services to primary care [61, 62]. Currently, the best available evidence in the field is a rapid review from 2021 by Blasi et al. [63] that identified practices for transitioning stable patients from outpatient mental health services to primary care, and a systematic review from 2006 by Gunn et al. [64] that assessed the effects of chronic illness management approaches for patients with depression in primary care. The rapid review by Blasi et al. [63] included 11 articles representing six categories of transition practices, with patient engagement as the most commonly described transition practice, followed by shared treatment planning, assessment of recovery and stability, care coordination, follow-up and support, and medication management. However, the review did not conclude about best practices or the importance of specific transition processes or strategies, including interventions that promote recovery for patients transitioning. In addition, the authors may have missed some relevant articles due to the rapid review timeline for literature searching and study selection [63]. The systematic review by Gunn et al. [64] found that system-level interventions in primary care can lead to a modest increase in recovery from depression. Yet, the quality of the evidence was poor and ten of the 11 randomized controlled trials included in the review (91%) were from the United States of America [64]. Thus, the authors concluded that possibly the findings in the reviews were likely not applicable to countries with strong primary care systems. Of note, the scope of the review was not recovery after discharge from mental health services. In addition, neither observational nor qualitative studies were included in review, limiting the review's ability to provide a comprehensive overview of the field of recovery from MDD [64]. Lastly, much research on recovery from MDD has been conducted since 2006, making an updated review relevant [65, 66]. Nevertheless, it is plausible that shared care models in treatment of MDD between outpatient mental health services and primary care may improve recovery [67]. Therefore, we believe that a scoping review on this field will be valuable to identify knowledge gaps due to its connection with and to inform an ongoing co-design development project that we will describe briefly below.

## The scoping review informs a co-design process of a complex intervention

Given the high rates of relapse and residual symptoms for patients with MDD following transitioning from outpatient mental health services to primary care, new strategies to promote continued recovery are required. A promising method is to develop an intervention that promotes continued recovery from MDD for patients transitioning. This scoping review is one part of an ongoing stakeholder co-design project (S1 Appendix) located in the Capital Region of Denmark. In the present review, stakeholders are involved in the design and conduct of the review. Other activities involved in the co-design process include individual interviews, focus groups, and workshops with stakeholders. The overall aim of the co-design project is to develop a complex intervention that promotes recovery from MDD for patients transitioning (S1 Appendix) from outpatient mental health services to primary care. Following the development of the intervention, we plan to test the intervention over a series of feasibility studies [68–70].

## Objective

The proposed scoping review aims to systematically scope, map and identify the evidence and knowledge gaps on interventions that aims to promote recovery from MDD for patients transitioning from outpatient mental health services to primary care.

Research questions (RQs):

- *RQ1*: *What characterizes studies conducted in the area in terms of settings, aims and methods*?

- *RQ2*: *How do studies define, measure, and report recovery from depression*?

- *RQ3*: *What is the content, implementation, and the most important contextual elements of the interventions in the identified studies*?

- *RQ4*: *What are the findings of the identified studies as to what promotes recovery from depression and which aspects promote the process*?

Hypothesis: We expect to find few studies that aim to promote recovery from MDD for patients transitioning from outpatient mental health services to primary care. In addition, we assume that studies use a diversity of designs and methods to assess the effects of recovery interventions. In recent years, the concept of recovery has become increasingly important in the mental health field. Therefore, we expect to find that definitions of the concept of recovery are very different across studies in the area. Finally, we expect to find different types of interventions that we are able to describe in terms of their content, implementation strategies and contextual factors important for promoting recovery from MDD.

## Materials and methods

Scoping reviews are methodologically rigorous in their approach to examining the extent, range, and nature of research activity in a particular field. The methodology is particularly useful for identifying and synthesizing the best available evidence that spans a vast conceptual and methodological range in the health disciplines [71–75], as is the case within this research area.

The first framework for conducting a scoping review was proposed by Arksey and O'Malley [71]. Extensions of this framework were later provided by Levac et al. [72]. These initial attempts have guided many researchers, but a lack of methodological clarity continues to exist. In response to ongoing concerns about the scoping review methodology, the Joanna Briggs Institute (JBI) guidance for scoping reviews was developed by a working group of methodological experts and first published in 2015 [73], and updated in 2017, 2020 [74, 76] and latest in 2022 [77].

This proposed scoping review will follow the latest methodological guidance by the JBI [77] in tandem with the Preferred Reporting Items for Systematic reviews and Meta-Analysis— extension for Scoping Reviews (PRISMA-ScR) checklist [78] (S2 Appendix).

### Patient and public involvement

During protocol development, we used the TRANSFER approach [79] to involve relevant stakeholders in discussions about the scoping reviews' aims and methods, aiming to promote relevance and transferability of the reviews' findings. We included a diverse set of stakeholders over a series of meetings to gain perspectives from researchers from a) general practice, b) mental health services, and c) social medicine. We also conducted interviews with patients with MDD and a focus group with job consultants from the Municipality of Copenhagen to include their perceptions.

### Protocol and registration

This scoping review protocol is novel, i.e., not based on updates from previous review(s). A pre-print has been registered at the medRxiv preprint server for health sciences (doi.org/10.1101/2022.10.06.22280499). In case the conduct outlined in this protocol changes substantially during the review process, we will update the protocol in the medRxiv preprint server accordingly and report deviations from the protocol in the final publication(s).

**Table 1. Eligibility criteria.**

| PICOS | Inclusion | Exclusion |
|---|---|---|
| Population | Adults (18 years of age or older) with the primary diagnosis MDD (as diagnosed using any recognized diagnostic criteria, e.g., DSM-IV or ICD-10, S1 Appendix). We will include patients with co-occurring disorders if they have a primary diagnosis of MDD. | Studies with exclusively elderly people (65 years of age or older), psychotic depression, depression as part of bipolar disorder, or people suffering exclusively from postpartum depression. Co-occurring alcohol or drug abuse, personality, phobia, or anxiety disorders are not exclusion criteria. |
| Intervention | We will include studies investigating any type of intervention e.g., simple, multicomponent, or complex interventions that aim to promote recovery from depression for patients transitioning from outpatient mental health services to primary care. This definition includes interventions that are both pharmacological and non-pharmacological, which can be delivered via the Internet, a website, a mobile-setting, in-person, or a mix thereof. | |
| Comparator | At this stage, any comparator will be included. In comparator studies, the control group can both receive treatment as usual, a placebo, an active ingrediency or alternative interventions. | *The criteria for comparator do not apply to qualitative studies.* |
| Outcome | Improvement in recovery from MDD. | *The criteria for outcome do not apply to qualitative studies.* |
| Setting | Patients must be in the transitioning setting from outpatient mental health services to primary care. This includes studies in which patients are nearing the end of their outpatient treatment in a mental healthcare setting, or patients who are being treated in primary care—we will only include studies concerning patients who are being treated in primary care if patients previously have been treated in an outpatient mental health service. | Patients who have not previously been treated in an outpatient mental health service, or patients who are recruited from an inpatient mental health service. |
| | We will include studies in which patients either have been or have not been hospitalized in an inpatient mental health service before their treatment course in an outpatient mental health service. | |

**Additional limits:** No limits on publication date, language, country, or gender, and no restrictions on the type of study design. Both qualitative, quantitative, and mixed-methods studies are included. Articles without full text available will not be included.

## Eligibility criteria

The following eligibility criteria (Table 1) guide the decision to in- or exclude studies identified for review. These are structured according to the PICOS acronym (population, intervention, comparator, outcome, and setting).

## Information sources and search

The literature search was developed in collaboration with an information specialist with feedback from the stakeholders that were included via the TRANSFER [79] approach in discussions regarding the eligibility criteria (PICOS elements) for the review outlined above. We have searched the electronic databases of Medline via PubMed, PsycINFO, CINAHL, and Sociological Abstracts. The search strategy included both text words and Medical Subject Headings (MeSh)/Thesaurus headings terms. Before performing the search strategy, we searched for ongoing or completed scoping or systematic reviews in the area on Cochrane Library, Google Scholar, and the PROSPERO register to make sure there were not already relevant reviews in the area.

The search strategy for PubMed is available in S3 Appendix. All databases were searched from 20 January 2022 to 29 March 2022.

Reference lists of included studies will be examined, i.e., backward citation tracing, to identify relevant studies potentially missed by the search strategy. Vice versa, we will do forward citation tracing of all included studies via Web of Science. The database searches will be re-run just before the final analysis is conducted to include the most recent evidence.

## Selection of sources of evidence

Results from the literature were exported from databases to the Covidence (https://www.covidence.org/) reference management software system. Duplication of database search results was removed used using EndNote 20 reference management software. Before the start of the review, all screeners were trained to use the Covidence system and received education about the content area, i.e., depression and recovery.

Relevant studies were screened through a two-step process for examining titles and abstracts and then full texts. The review team consists of three unblinded screeners. Two independent screeners completed the initial title and abstract screening for all studies retrieved by the search strategy. A third screener reviewed conflicts and resolved disagreements through discussion with the two other screeners. Over two months, these unblinded screeners (unable to see each other votes in Covidence until they have cast their own, and vice versa, and they will not be blinded to the authors and journals) have screened 4605 titles and abstracts independently. Three screeners reviewed at least 1600 titles and abstracts each. Currently, we are in step two, the full-text screening. Here, two independent screeners will review the full text of potentially eligible articles. Disagreements between screeners during full-text screening will be resolved by discussion or, if needed, by consulting a third screener. If there is more than one article from the same study, the most updated data will be extracted. If information is missing or clarification of data is required, authors will be contacted via e-mail.

Overall reasons for the inclusion/exclusion of studies will be documented and reported in a PRISMA flowchart [80] in the final article reporting the findings from the review process.

## Data charting process

The preliminary charting table (Table 2) guides data extraction (charting). Design of the table was guided by/-inspired by the newest JBI guideline [77] and further developed for this scoping review in line with the review's objectives and research questions in collaboration with stakeholders included via the TRANSFER approach [79]. Two review authors will independently extract data from included studies into a Microsoft Excel sheet organized in columns corresponding to the items in the table. The review authors will agree on revisions to the charting table as needed in an iterative process [81]. To ensure clarity and consistency between the screeners' data extraction, and before initiating the full-text article selection process, we will pilot test the data extraction process on a subset of potentially eligible full-text articles. Review authors will resolve disagreements by discussion, and a third review author will adjudicate unresolved disagreements.

## Data items

We will extract data as shown in Table 2.

## Critical appraisal of individual sources of evidence

Since this is a scoping review, we will not conduct a quality appraisal of included studies, which is consistent with the framework proposed by the JBI methodology for scoping reviews [74, 76]. Still, the independent data extraction by two review authors with comparison of the extracted data and discussion of any disagreements should lead to a high reliability of the extracted data, i.e., that the data extracted represents the findings and characteristics of the studies included for review. In addition, the first author (ASA) iteratively compares the extracted data with the publications from the included studies on a case-by-case basis as a second precaution against erroneous data extraction.

**Table 2. Preliminary charting table.**

| Item | Description |
|------|-------------|
| Author(s) | |
| Title | |
| Year of publication | |
| Journal | |
| Country | *By country* |
| | *By income category (high-income, middle-income and low-income countries)* |
| Study design | *Systematic review, randomized controlled trial, qualitative studies etc.* |
| Aims/objectives of the study | |
| Study population | *Sample size, i.e., number of participants, gender, age* |
| Methodology/methods | *Quantitative, qualitative* |
| Intervention characteristics | *Type: Specify the type of interventions on which the study focuses* |
| | *Delivery of interventions: Describe how and by whom the intervention is delivered* |
| | *Length and intensity of the interventions: Describe how long the intervention is delivered, the setting, its intensity, frequency, and comparator (if available)* |
| Setting of the intervention(s) | *Specify if the study focuses on interventions delivered in e.g., primary care or community-based settings* |
| Key findings relating to the review question | *Acceptability of the intervention from care providers and patients, experiences with receiving or delivering the intervention, costs, any outcome part of recovery, and fidelity to the intervention* |
| Facilitators for recovery | *Describe the factors that support or enable the implementation of the intervention reported in the study* |
| Barriers for recovery | *Describe the factors that inhibit the implementation of the intervention reported in the study* |

## Synthesis of results

According to the JBI methodology [74, 76] for scoping reviews, the quantitative results extracted from included studies will be analyzed with descriptive statistics with visual representations of the data where possible, e.g., mapping the extracted data in a diagrammatic, tabular, or descriptive format. Qualitative findings from studies will be analyzed from a thematic perspective and, depending on the results, described regarding for example active ingredients, patient satisfaction, and barriers and facilitators for implementation.

The results will be classified under main conceptual categories, such as: "intervention type", "duration of intervention", "facilitators/barriers", "aims", "methodology adopted", "key findings" (evidence established), and "gaps in the research field". For each category reported, a clear explanation will be provided.

## Discussion

This scoping review constitutes the first step of a larger research project aiming to develop a complex intervention to promote recovery from MDD in patients transitioning from outpatient mental health services to primary care. The chosen methodology is based on the use of publicly available information and does not require ethical approval.

### Strengths and limitations

To our knowledge, this will be the first scoping review that systematically scope, map, and identify the evidence and knowledge gaps on interventions that promote recovery from MDD for patients transitioning from outpatient mental health services to primary care.

Scoping reviews form a method used to map evidence across a range of study designs in an area, with the aim of informing future research practice, programs, and policy. However, no universal agreement exists on methodological steps, and therefore several guidelines have developed methods for conducting scoping reviews. We will conduct the review in accordance with the PRISMA-ScR in tandem with the Joanna Briggs Institute's (JBI) framework. Concerning data extraction, we will follow the template for intervention description and replication (TIDieR) checklist and guide to ensure a systematic extraction of data. Another strength of this scoping review is the involvement of stakeholders guided by the TRANSFER guide [79], which promotes integration of different perspectives on the aim, design, and methods of the scoping review from health professionals representing psychiatry, social medicine and primary care, and both clinicians, researchers and patients are involved. The goal of this process is to promote an evidence synthesis that is relevant to key stakeholders, e.g., facilitating a link between research and real-world clinical practice.

However, the scoping review methodology comes with important limitations like any type of review. Scoping reviews focus on mapping the breath and range of the literature rather than the depth, i.e., the validity of findings. Therefore, we will present an overview of the field rather than an evidence synthesis of the probable effect of various types of recovery-interventions. In addition, scoping reviews focus on describing knowledge gaps in the literature rather than contributing with new knowledge. As is typical for scoping reviews, we do not assess study quality or bias, nor will we provide a systematic assessment of the external validity of the evidence, i.e., a GRADE rating. Instead, we will outline the key characteristics of the best-available evidence in the area and comment of the applicability of the evidence in various settings.

Another limitation concerns the expected heterogeneity of the evidence at hand. We expect to find studies that include different study populations, e.g., some participants are included from general practice and others from outpatient mental health services. If study populations in primary care and outpatient mental health services are too different, it would not be meaningful to compare results from these two different settings. To mitigate these issues, we will only include studies where participants have been treated in mental health services, which theoretically should prevent that we conflate study populations with different degrees of depression. Still, we cannot prevent that study populations will differ concerning their average age, social class, degree of depression, comorbidities etc., and therefore we will try to highlight these differences when synthesizing results from studies in the final review.

## Supporting information

**S1 Appendix. List of abbreviations / concepts.**
(PDF)

**S2 Appendix. Preferred Reporting Items for Systematic Reviews and Meta-Analyses extension for Scoping Reviews (PRISMA-ScR) checklist.**
(PDF)

**S3 Appendix. Search strategy.**
(PDF)

## Acknowledgments

Depression, recovery, transitioning, mental health services, primary care. Data sharing not applicable as no datasets generated and/or analyzed for this study.

## Author Contributions

**Data curation:** Anne Sofie Aggestrup, Frederik Martiny, Annette Sofie Davidsen, Klaus Martiny.

**Investigation:** Anne Sofie Aggestrup, Frederik Martiny, Annette Sofie Davidsen, Klaus Martiny.

**Methodology:** Anne Sofie Aggestrup, Frederik Martiny, Maria Faurholt-Jepsen, Annette Sofie Davidsen, Klaus Martiny.

**Project administration:** Anne Sofie Aggestrup.

**Resources:** Anne Sofie Aggestrup, Frederik Martiny, Maria Faurholt-Jepsen, Morten Hvenegaard, Robin Christensen, Annette Sofie Davidsen, Klaus Martiny.

**Supervision:** Frederik Martiny, Maria Faurholt-Jepsen, Morten Hvenegaard, Robin Christensen, Annette Sofie Davidsen, Klaus Martiny.

**Validation:** Frederik Martiny, Annette Sofie Davidsen.

**Visualization:** Anne Sofie Aggestrup.

**Writing – original draft:** Anne Sofie Aggestrup.

**Writing – review & editing:** Frederik Martiny, Maria Faurholt-Jepsen, Morten Hvenegaard, Robin Christensen, Annette Sofie Davidsen, Klaus Martiny.

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
