## [Decision Letter · Decision Letter 0]

7 Jul 2023

PONE-D-23-02223Interventions Promoting Recovery from Depression for Patients Transitioning from Outpatient Mental Health Services to Primary Care: Protocol for a Scoping ReviewPLOS ONE

Dear,

Thank you for submitting your manuscript to PLOS ONE. After careful consideration, we feel that it has merit but does not fully meet PLOS ONE’s publication criteria as it currently stands. Therefore, we invite you to submit a revised version of the manuscript that addresses the points raised during the review process.

1-Can you proofread the documents for minor language and grammar issues?

2-Could you elaborate your research question. Please provide more RQ and what is your hypothesis.

3-How would you validate the results?

4- The current given strength and limitation of protocol is not in depth and did not address all bias or outcome in context to strength of study. Write it in details.

5-Please specify on how you disseminate your finding. The generalized statement is not the right method.

We look forward to receiving your revised manuscript.

Kind regards,

Muhammad Shahzad Aslam, Ph.D.,M.Phil., Pharm-D

Academic Editor

PLOS ONE

Journal Requirements:

Additional Editor Comments (if provided):

1-Can you proofread the documents for minor language and grammar issues?

2-Could you elaborate your research question. Please provide more RQ and what is your hypothesis.

3-How would you validate the results?

4- The current given strength and limitation of protocol is not in depth and did not address all bias or outcome in context to strength of study. Write it in details.

5-Please specify on how you disseminate your finding. The generalized statement is not the right method.

Reviewers' comments:

Reviewer's Responses to Questions

**Comments to the Author**

1. Does the manuscript provide a valid rationale for the proposed study, with clearly identified and justified research questions?

Reviewer #1: Yes

Reviewer #2: Yes

2. Is the protocol technically sound and planned in a manner that will lead to a meaningful outcome and allow testing the stated hypotheses?

Reviewer #1: Yes

Reviewer #2: Yes

3. Is the methodology feasible and described in sufficient detail to allow the work to be replicable?

Reviewer #1: Yes

Reviewer #2: Yes

4. Have the authors described where all data underlying the findings will be made available when the study is complete?

Reviewer #1: Yes

Reviewer #2: Yes

5. Is the manuscript presented in an intelligible fashion and written in standard English?

Reviewer #1: Yes

Reviewer #2: Yes

6. Review Comments to the Author

You may also provide optional suggestions and comments to authors that they might find helpful in planning their study.

Reviewer #1: A well-thought out, practicable and meaningfully pragmatic protocol for a scoping review of interventions promoting recovery from depression for patients transitioning from outpatient mental health services to Primary Care.

Reviewer #2: The study as defined and written is fit, adequate and commendable to reviewer appraisal for acceptance.

However, the proposed study’s exclusion criteria do not include depression with commonly co-occurring disorders such as substance use disorder(s) or psychotic depressions or depression with psychosis or sequelae of psychosis, whose omission proven an unusual exception given an expectable empirical standard and the fact that not just drug abuse but thought disorders like psychosis affecting depression presumably entail highly unique barriers of care across system tiers or levels, potentially rendering dubious shared putative empirical load when transitioning care services.

More urgently perhaps, I would seek to know if the eventual quantitative analyses will adequately account for various temporal and epochal considerations since these should be outlined a priori, such as whether studies will be assessed for homogeneity based on lengths of outpatient exposures in the study populations (since for the present study in particular this will not be an un-impactful statistical control), and more pressingly I wonder if it is entirely appropriate to include previous inpatient groups due to diagnostic conflation, urgent care diagnostic thresholds compared to general population diagnostic thresholds and the much higher service needs and resultant statistical bias of inpatient populations.

7. PLOS authors have the option to publish the peer review history of their article (what does this mean?). If published, this will include your full peer review and any attached files.

Reviewer #1: **Yes: **Dr Sarah Markham

Reviewer #2: **Yes: **Paul-Andre Betito

---

## [Author Response · Author response to Decision Letter 0]

30 Aug 2023

Dear Muhammad Shahzad Aslam, Ph.D., M.Phil., Pharm-D

Thank you for your feedback and for providing us with the opportunity for revising our manuscript entitled "Interventions Promoting Recovery from Depression for Patients Transitioning from Outpatient Mental Health Services to Primary Care: Protocol for a Scoping Review”. We thank the reviewers for their thorough evaluation of our manuscript. Please find a point-to-point reply to yours and the reviewer’s comments in the three tables below. The changes made to the manuscript are marked with track changes. 

Academic Editor

Comments from editor: 1. Can you proofread the documents for minor language and grammar issues? 

Answers from authors: Thank you for your suggestions, we have proofread the whole document for language and grammar issues. 

Comments from editor: 2. Could you elaborate your research question. Please provide more RQ and what is your hypothesis. 

Answers from authors: We agree that the research questions could be clearer, therefore we have changed the research questions to the following (on p. 6 in the manuscript): 

Research questions (RQs): 

• RQ1: What characterizes studies conducted in the area in terms of settings, aims and methods? 

• RQ2: How do studies define, measure, and report recovery from depression? 

• RQ3: What is the content, implementation, and the most important contextual elements of the interventions in the identified studies?

• RQ4: What are the findings of the identified studies as to what promotes recovery from depression and which aspects promote the process?

Hypothesis: We expect to find few studies that aim to promote recovery from MDD for patients transitioning from outpatient mental health services to primary care. In addition, we assume that studies use a diversity of designs and methods to assess the effects of recovery interventions. In recent years, the concept of recovery has become increasingly important in the mental health field. Therefore, we expect to find that definitions of the concept of recovery are very different across studies in the area. Finally, we expect to find different types of interventions that we are able to describe in terms of their content, implementation strategies and contextual factors important for promoting recovery from MDD.

Comments from editor: 3. How would you validate the results? 

Answers from authors: Thank you for your question. For each included study, two review authors have independently extracted data into a Microsoft Excel sheet. Thereafter, the two data extracts were compared to ensure reliable data extraction. In addition, the first author (ASA) compared the extracted data with the written text in the publications from the included studies on a case-by-case basis as a second precaution to ensure that data represents the actual findings from included studies. However, as this is a scoping review, which does not include a quality appraisal of included studies, as is the norm, we cannot ensure that results from studies are valid. 

To make these distinctions clearer, we have added the following with track changes to the manuscript on p. 10 (section: Critical appraisal of individual sources of evidence): 

Critical appraisal of individual sources of evidence

Since this is a scoping review, we will not conduct a quality appraisal of included studies, which is consistent with the framework proposed by the JBI methodology for scoping reviews [74, 76]. Still, the independent data extraction by two reviewers followed by comparison and discussion of any disagreements should lead to a high reliability of the extracted data, i.e., that the data extracted represents the findings and characteristics of the studies included for review. In addition, first author (ASA) iteratively compared the extracted data with the publications from the included studies on a case-by-case basis as a second precaution against erroneous data extraction.

Comments from editor: 4. The current given strength and limitation of protocol is not in depth and did not address all bias or outcome in context to strength of study. Write it in detail. 

Answers from authors: Thank you for bringing our attention to this shortcoming in our manuscript. We have now added a more detailed outline of the strength and limitation of our scoping review protocol and added a discussion section. 

In the manuscript p. 11 (section: Discussion) we have added the following: 

DISCUSSION

This scoping review constitutes the first step of a larger research project aiming to develop a complex intervention to promote recovery from MDD in patients transitioning from outpatient mental health services to primary care. The chosen methodology is based on the use of publicly available information and does not require ethical approval. 

Strengths and limitations

To our knowledge, this will be the first scoping review that systematically scope, map, and identify the evidence and knowledge gaps on interventions that promote recovery from MDD for patients transitioning from outpatient mental health services to primary care.

Scoping reviews form a method used to map evidence across a range of study designs in an area, with the aim of informing future research practice, programs, and policy. However, no universal agreement exists on methodological steps, and therefore several guidelines have developed methods for conducting scoping reviews. We will conduct the review in accordance with the PRISMA-ScR in tandem with the Joanna Briggs Institute’s (JBI) framework. Concerning data extraction, we will follow the template for intervention description and replication (TIDieR) checklist and guide to ensure a systematic extraction of data. Another strength of this scoping review is the involvement of stakeholders guided by the TRANSFER guide [79], which promotes integration of different perspectives on the aim, design, and methods of the scoping review from health professionals representing psychiatry, social medicine and primary care, and both clinicians, researchers and patients are involved. The goal of this process is to promote an evidence synthesis that is relevant to key stakeholders, e.g., facilitating a link between research and real-world clinical practice. 

However, the scoping review methodology comes with important limitations like any type of review. Scoping reviews focus on mapping the breath and range of the literature rather than the depth, i.e., the validity of findings. Therefore, we will present an overview of the field rather than an evidence synthesis of the probable effect of various types of recovery-interventions. In addition, scoping reviews focus on describing knowledge gaps in the literature rather than contributing with new knowledge. As is typical for scoping reviews, we do not assess study quality or bias, nor will we provide a systematic assessment of the external validity of the evidence, i.e., a GRADE rating. Instead, we will outline the key characteristics of the best-available evidence in the area and comment of the applicability of the evidence in various settings. 

Another limitation concerns the expected heterogeneity of the evidence at hand. We expect to find studies that include different study populations, e.g., some participants are included from general practice and others from outpatient mental health services. If study populations in primary care and outpatient mental health services are too different, it would not be meaningful to compare results from these two different settings. To mitigate these issues, we will only include studies where participants have been treated in mental health services, which theoretically should prevent that we conflate study populations with different degrees of depression. Still, we cannot prevent that study populations will differ concerning their average age, social class, degree of depression, comorbidities etc., and therefore we will try to highlight these differences when synthesizing results from studies in the final review. 

Comments from editor: 5. Please specify on how you disseminate your finding. The generalized statement is not the right method.

Answers from authors: We agree that a clear strategy on how to disseminate findings from the review at the outset is important. Therefore, we have added a more detailed plan for how we expect to do this. 

In the manuscript p. 12 (section: Dissemination of findings), we have added the following: 

DISSEMINATION OF FINDINGS

We expect that the reviews findings will be of interest both to researchers and to professionals who are interested in recovery-interventions for patients transitioning from outpatient mental health services to primary care. Therefore, we plan to disseminate our results in a peer-reviewed international scientific journal, at national and international conferences, and to share them with relevant local and national authorities, researchers, and other interested stakeholders, e.g., at oral presentations and poster presentations provided at various research institutions, healthcare clinics and municipalities. In addition, we will publish our findings via other communication channels, e.g., online on our institution's website, via social media platforms, traditional media, and established science communication forums, e.g., ResearchGate. 

Reviewer #1

Comments from reviewer #1: A well-thought out, practicable and meaningfully pragmatic protocol for a scoping review of interventions promoting recovery from depression for patients transitioning from outpatient mental health services to Primary Care. 

Answers from authors: Thank you very much for your positive response. 

Reviewer #2

Comments from reviewer #2: The study as defined and written is fit, adequate and commendable to reviewer appraisal for acceptance. However, the proposed study’s exclusion criteria do not include depression with commonly co-occurring disorders such as substance use disorder(s) or psychotic depressions or depression with psychosis or sequelae of psychosis, whose omission proven an unusual exception given an expectable empirical standard and the fact that not just drug abuse but thought disorders like psychosis affecting depression presumably entail highly unique barriers of care across system tiers or levels, potentially rendering dubious shared putative empirical load when transitioning care services.

Answers from authors: Thank you for your positive response and suggestions. Many trials often include highly selected patients, i.e., ideal trials, to study efficacy rather than effectiveness, i.e., pragmatic trials, which leads to high internal validity at the cost of external validity, e.g., major differences between selected groups of patients in studies and real-world patients such as relevant treatment effect modifiers, like comorbidity. To provide evidence that is applicable in the real-world, we chose to include both ideal and pragmatic trials where the latter often studies less selected groups of patients with co-occurring disorders to generate the real-world in routine clinical practice as done in pragmatic clinical trials. However, we do acknowledge that psychotic patients are significantly different from non-psychotic patients with MDD, needing distinctly different treatments. Therefore, we have added this as an exclusion criterion. 

In the manuscript p. 7, table 1, we have added the following: 

Eligibility criteria 

Inclusion criteria: Adults (18 years of age or older) with the primary diagnosis MDD (as diagnosed using any recognized diagnostic criteria, e.g., DSM-IV or ICD-10, supplementary file A). We will include patients with co-occurring disorders if they have a primary diagnosis of MDD.

Exclusion criteria: Studies with exclusively elderly people (65 years of age or older), psychotic depression, depression as part of bipolar disorder, or people suffering exclusively from postpartum depression. Co-occurring alcohol or drug abuse, personality, phobia, or anxiety disorders are not exclusion criteria.

Comments from reviewer #2: More urgently perhaps, I would seek to know if the eventual quantitative analyses will adequately account for various temporal and epochal considerations since these should be outlined a priori, such as whether studies will be assessed for homogeneity based on lengths of outpatient exposures in the study populations (since for the present study in particular this will not be an un-impactful statistical control), and more pressingly I wonder if it is entirely appropriate to include previous inpatient groups due to diagnostic conflation, urgent care diagnostic thresholds compared to general population diagnostic thresholds and the much higher service needs and resultant statistical bias of inpatient populations. 

Answers from authors: Thank you for your suggestions. As this is a scoping review that aims to provide an overview of the field rather than quantitative assessment of e.g., treatment effects, we will not include statistical analyses. Please view our reply to editor (item 4), in addition, we have added your suggestions to our discussion section – strengths and limitations - in the manuscript, p. 11-12: 

Strengths and limitations

In addition, scoping reviews focus on describing knowledge gaps in the literature rather than contributing new knowledge. As is typical for scoping reviews, we do not assess study quality or bias, nor will we provide a systematic assessment of the external validity of the evidence, i.e., a GRADE rating. Instead, we will outline the key characteristics of the best-available evidence in the area and comment of the applicability of the evidence in various settings. 

Another limitation concerns the expected heterogeneity of the evidence at hand. We expect to find studies that include different study populations, e.g., some participants are included from general practice and others from outpatient mental health services. If study populations in primary care and outpatient mental health services are too different, it would not be meaningful to compare results from these two different settings. To mitigate these issues, we will only include studies where participants have been treated in mental health services, which theoretically should prevent that we conflate study populations with different degrees of depression. Still, we cannot prevent that study populations will differ concerning their average age, social class, degree of depression, comorbidities etc., and therefore we will try to highlight these differences when synthesizing results from studies in the final review.

---

## [Editor Report · Decision Letter 1]

1 Sep 2023

Interventions Promoting Recovery from Depression for Patients Transitioning from Outpatient Mental Health Services to Primary Care: Protocol for a Scoping Review

PONE-D-23-02223R1

Dear,

We’re pleased to inform you that your manuscript has been judged scientifically suitable for publication and will be formally accepted for publication once it meets all outstanding technical requirements.

Kind regards,

Muhammad Shahzad Aslam, Ph.D.,M.Phil., Pharm-D

Academic Editor

PLOS ONE
---

## [Editor Report · Acceptance letter]

7 Sep 2023

PONE-D-23-02223R1 

Interventions Promoting Recovery from Depression for Patients Transitioning from Outpatient Mental Health Services to Primary Care: Protocol for a Scoping Review 

Dear Dr. Aggestrup:

I'm pleased to inform you that your manuscript has been deemed suitable for publication in PLOS ONE. Congratulations! Your manuscript is now with our production department. 

Kind regards, 

on behalf of

Dr. Muhammad Shahzad Aslam 

Academic Editor

PLOS ONE